# Temperature-dependent dual-mode thermal management device with net zero energy for year-round energy saving

Quan Zhang[1,3], Yiwen Lv[1,3], Yufeng Wang[1], Shixiong Yu[1], Chenxi Li[2], Rujun Ma [1] ✉ & Yongsheng Chen [2] ✉

Reducing needs for heating and cooling from fossil energy is one of the biggest challenges, which demand accounts for almost half of global energy consumption, consequently resulting in complicated climatic and environmental issues. Herein, we demonstrate a high-performance, intelligently auto-switched and zero-energy dual-mode radiative thermal management device. By perceiving temperature to spontaneously modulate electromagnetic characteristics itself, the device achieves ~859.8 W m$^{-2}$ of average heating power (~91% of solar-thermal conversion efficiency) in cold and ~126.0 W m$^{-2}$ of average cooling power in hot, without any external energy consumption during the whole process. Such a scalable, cost-effective device could realize two-way temperature control around comfortable temperature zone of human living. A practical demonstration shows that the temperature fluctuation is reduced by ~21 K, compared with copper plate. Numerical prediction indicates that this real zero-energy dual-mode thermal management device has a huge potential for year-round energy saving around the world and provides a feasible solution to realize the goal of Net Zero Carbon 2050.

Thermal management plays an important role in human activities, from million cubic meters of human-made structure[1] to micro- and nano-scale integrated circuits[2], and from spacecraft flying in outer space[3] to deep-sea manned submersible[4]. Various thermal management technologies have been developed according to different requirements[5–7]. However, most of them achieve high-performance temperature control at the cost of energy consumption, eventually fossil energy. Reports have pointed out that the global total primary energy demand is close to a 15-billion-tonne oil equivalent in 2019[8], and nearly 50% of energy consumption is merely used for daily heating and cooling[9]. This particularly makes the increasing energy crisis continue to worsen. Meanwhile, with the rapid increase of greenhouse gases produced by the combustion of fossil fuels, extreme weather, such as intense heat and severe cold, has frequently increasingly

occurred worldwide in recent years[10]. Therefore, it is particularly important and imperative to develop various feasible high-performance thermal management technologies with low or even zero energy consumption, which is able to reduce fossil energy demand and further emission of greenhouse gases.

Radiative thermal management is regarded as a promising platform for heating and cooling without external energy consumption, attracting more and more attention[11]. The most challenging issue to realize this goal is to optimize the unique electromagnetic spectrum of the thermal management materials, maximizing the utilization of both the inexhaustible radiative heat source (i.e., the sun, ~5800 K) and cool source (i.e., outer space, ~3 K) in nature. More specially, for ideal solar heating, the materials should have high absorptivity in the wavelength range of 0.2–2.5 μm and

[1]School of Materials Science and Engineering, National Institute for Advanced Materials, Nankai University, Tongyan Road 38, Tianjin 300350, P. R. China.
[2]State Key Laboratory and Institute of Elemento-Organic Chemistry, Centre of Nanoscale Science and Technology and Key Laboratory of Functional Polymer Materials, College of Chemistry, Nankai University, Tianjin 300071, P. R. China. [3]These authors contributed equally: Quan Zhang, Yiwen Lv.
✉e-mail: malab@nankai.edu.cn; yschen99@nankai.edu.cn

low emissivity in the wavelength range >2.5 μm, determined by the sunlight spectrum and blackbody radiation law[12]. On the contrary, for ideal radiative cooling, especially in daytime sub-ambient radiative cooling, the materials are expected to efficiently reflect solar radiation (0.2–2.5 μm) and also have strong selective mid-infrared emission in the specific wavelength range of transparent atmospheric window (8–13 μm) (Fig. 1)[13]. Note a series of studies on solar heating and radiative cooling separately/independently have made great efforts to thoroughly understand the scientific mechanism and develop high-efficient materials[14–20]. Nevertheless, in the real world, almost all the ambient scenarios come with the challenge that the objects are located in a quite dynamic and variable environment, including the fluctuation in the aspects of space, time, day and season, temperature etc. It means that fixed solar heating or radiative cooling are both not completely suitable for the dynamic ambient. Taking solar heating as an example, unwanted heating will increase the energy consumption for cooling in the hot and even may offset the energy saving of heating in the cold. The same is true of radiative cooling. Therefore, for practical utilization, a thermal management system, being able to possess both above two opposite electromagnetic spectrums and automatically/intelligently switch to the right mode by responding to the dynamic ambient, is required.

Up to now, several approaches have been designed to dynamically regulate the spectral characteristics of dual-mode thermal management in the literature (Supplementary Table 1)[21–31]. However, there is more or less external energy cost for switching between heating and cooling modes, such as using mechanical energy or electrical energy. In other words, these designs are quasi-zero-energy dual-mode ther-

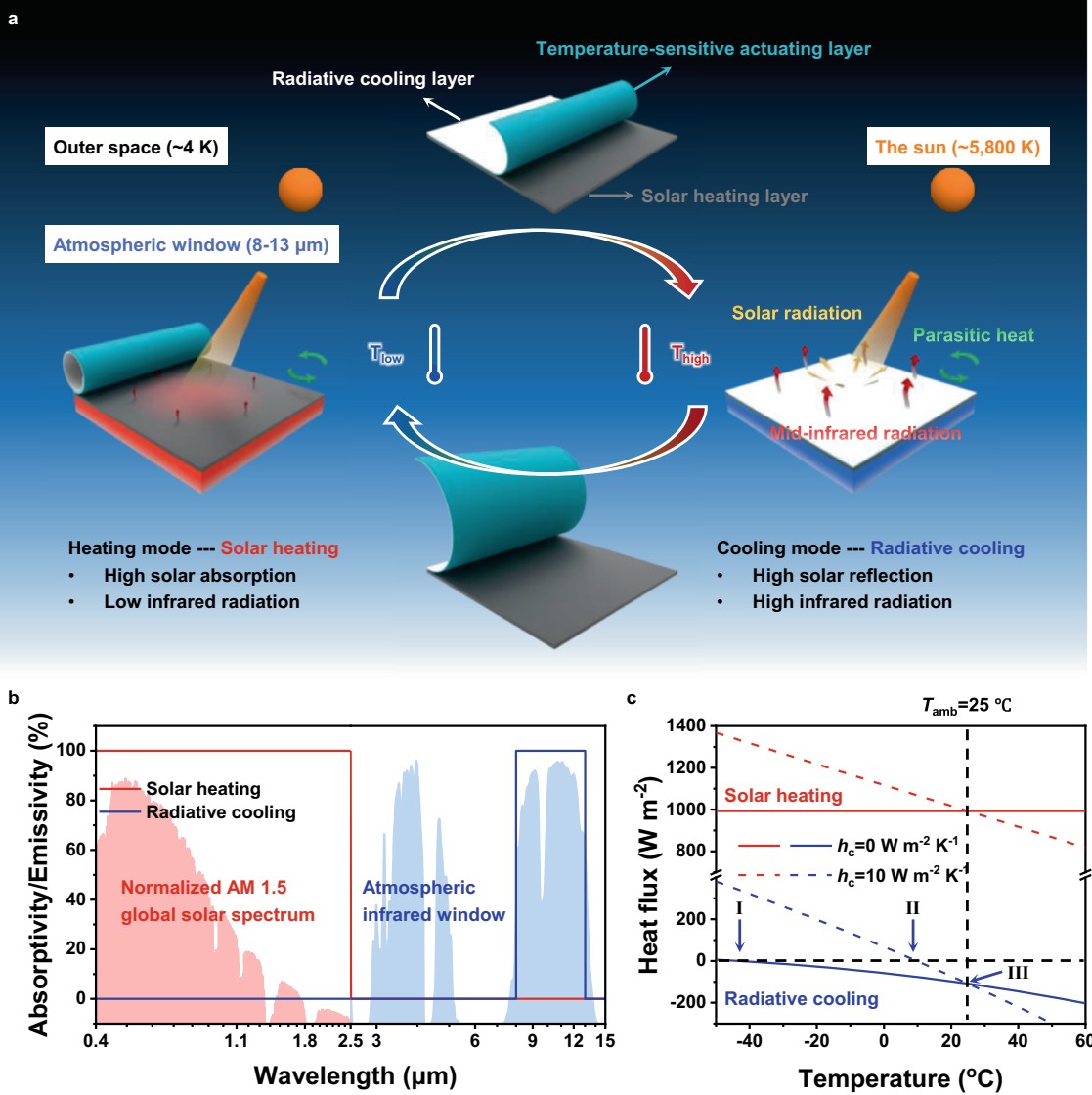

**Fig. 1 | Design principle of intelligent auto-switched and zero-energy dual-mode radiative thermal management device. a** Schematic illustration of the dual-mode radiative thermal management device switching between solar heating (left) and radiative cooling (right) with temperature. The dual-mode device (top) consists of three functional layers: radiative cooling layer, temperature-sensitive actuating layer, and solar heating layer (not to scale). **b** Absorptivity/emissivity spectrum of ideal solar heating (red line) and radiative cooling (blue line) materials. Normalized ASTM G173 Global solar spectrum (light-red area) and transparent infrared atmospheric window (US standard 1976, light-blue area) are plotted for reference. **c** Net heat flux as a function of the temperature of the ideal solar heating (red) and radiative cooling (blue) materials. Note that the heat flux is calculated by thermal balance relationship (Supplementary Note 1) based on the global solar spectrum (ASTM G173) and the typical transparent atmospheric window (US standard 1976). The steady-state temperature of the material is reached when net heat flux is zero. The thermal management power is the intersection corresponding to zero temperature difference between the material and the ambient, where positive heat flux and negative heat flux represent heating power and cooling power, respectively.

mal management, although no external energy is cost during solar heating and radiative cooling.

Here, we develop an intelligent auto-switched and zero-energy dual-mode thermal management device, which is able to spontaneously switch between heating and cooling modes by perceiving the ambient temperature. The zero-energy thermal management relies on the two high-selectivity but different electromagnetic properties corresponding to solar heating and radiative cooling layers, respectively. And the zero-energy switch between two different modes is realized by the automatic actuation of the device, from the mismatch of the shape between the radiative cooling layer and the actuating layer with temperature change. The results of the field test show that the device has an average solar heating power of ~859.8 W m$^{-2}$ (corresponding to ~91% solar-thermal conversion efficiency) and an average radiative cooling power of ~126.0 W m$^{-2}$, both of which are comparable with state-of-the-art solar heating and radiative cooling materials alone. Excellent thermal management performance and ability to automatically switch make the device enable to choose the right mode to achieve the best temperature control results. Numerical prediction reveals the great potential of this dual-mode device in terms of global energy saving. Such a zero-energy thermal management could contribute to the realization of the goal of Net Zero Carbon 2050.

## Results

### Concept of zero-energy intelligent dual-mode device

As shown in Fig. 1a, dual-mode thermal management device consists of three functional layers, which are in order as follows: radiative cooling layer, temperature-sensitive actuating layer and solar heating layer. The essence of zero-energy dual-mode radiative thermal management strategy is based on the transform of required different high-selectivity spectral characteristics in the temperature control system (Fig. 1b). When heating mode is required, the radiative cooling layer is coiled automatically to maximum the uncovered solar heating layer. Due to the high solar absorptivity and the low infrared emissivity of solar heating layer, most of the solar radiation is absorbed and converted into heat, and the heat loss from infrared radiation is suppressed to a minimum. For cooling mode, the automatic unfolded radiative cooling layer completely covers the solar heating layer, where high solar reflection of radiative cooling layer on the sunlight reduces solar absorption as much as possible, thereby avoiding increase of internal energy from solar radiation. Meanwhile, the high mid-infrared emission in the specific wavelength range (8–13 μm) directly transfers heat through the transparent atmospheric window into outer space by full power thermal radiation, reducing undesired input infrared radiation from air and surrounding environment. The steady-state temperature of dual-mode device is determined by the thermal balance relationship among four key components: the absorbed solar radiation from the sun ($P_{sun}$), the emitted heat by the device ($P_{device}$), the absorbed heat radiation from the atmosphere ($P_{atm}$), and the parasitic heat ($P_{parasitic}$) characterized by a heat-transfer coefficient ($h_c$) (Eq. (1) and

Supplementary Note 1)[14]. The net heat flux ($P_{net}$) is a function of the temperature of the device ($T_{device}$).

$$P_{net} = P_{sun} + P_{atm} - P_{device} - P_{parasitic} \tag{1}$$

$$P_{parasitic} = A_{device} h_c (T_{device} - T_{amb}) \tag{2}$$

Here, we fixed the ambient temperature ($T_{amb}$) to be 25 °C, and used the universal global solar spectrum (ASTM G173) and the typical atmospheric window (US standard 1976). When the net heat flux is zero, the steady-state temperature of the device is reached, and the thermal management power (negative represents cooling, positive represents heating) is the intersection corresponding to the temperature of the device equal to that of the ambient (Fig. 1c). The former is sensitive to parasitic heat. Taking cooling mode as an example, the steady-state temperature of the device is gradually close to the ambient temperature (from I to II) with increase of heat-transfer coefficient (from 0 to 10 W m$^{-2}$ K$^{-1}$). Different from steady-state temperature, radiative cooling power is independent from parasitic heat (III). This analysis is also suitable for heating mode.

The auto-switching mechanism is based on the spontaneous morphological adjustment of the dual-mode device responding to the ambient temperature change (Fig. 1a). The length of the actuating layer is sensitive to temperature, but the length of the radiative cooling layer is almost unchanged under the same conditions. When it is hot, the actuating layer shrinks. To eliminate the internal stress at the interface between the radiative cooling layer and actuating layer, the radiative cooling layer gradually unfolds until completely covering the solar heating layer for cooling. When it is cold, the actuating layer responds in the opposite way to expose the solar heating layer as much as possible. More importantly, the stimulus triggering the switch of thermal management modes is temperature, which is the physical quantity that determines the requirements of thermal management. This means that the dual-mode device is intelligent and can select an appropriate mode according to the ambient temperature, without any external energy consumption during the whole switching process.

We summarize that the successful realization of an intelligent and zero-energy dual-mode thermal management device requires three typical characteristics (Fig. 2): (a) The device should have high-selectivity electromagnetic spectrum in both heating and cooling modes to obtain dual-mode high thermal management performance. (b) The device has the ability to switch between heating and cooling modes by using the change in its own physical-chemical properties. This is a key factor to realize zero-energy thermal management. (c) The reversible auto-switch of thermal management mode should be triggered by temperature. Combining these three characteristics together would not only give the dual-mode device "intelligence" to choose an appropriate mode by perceiving the ambient automatically with zero-

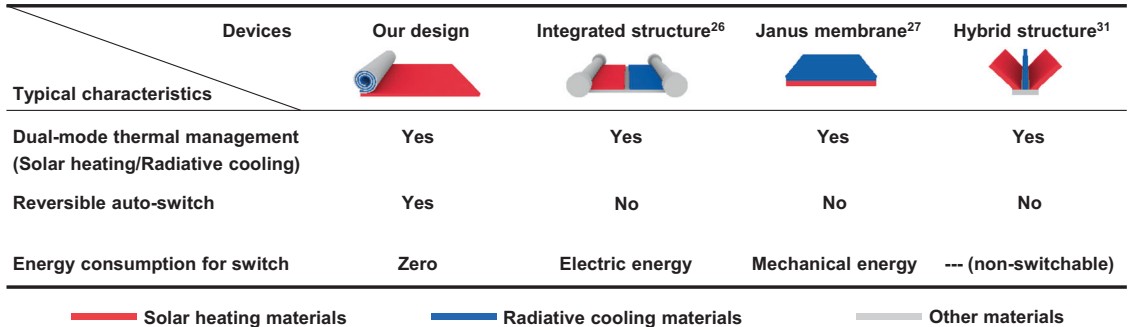

| Devices Typical characteristics | Our design | Integrated structure[26] | Janus membrane[27] | Hybrid structure[31] |
|---|---|---|---|---|
| Dual-mode thermal management (Solar heating/Radiative cooling) | Yes | Yes | Yes | Yes |
| Reversible auto-switch | Yes | No | No | No |
| Energy consumption for switch | Zero | Electric energy | Mechanical energy | --- (non-switchable) |

Solar heating materials Radiative cooling materials Other materials

**Fig. 2 | Feature comparison of the typical dual-mode thermal management devices.** Three criteria of the dual-mode device: dual-mode thermal management (solar heating/radiative cooling), reversible auto-switch, energy consumption for switch.

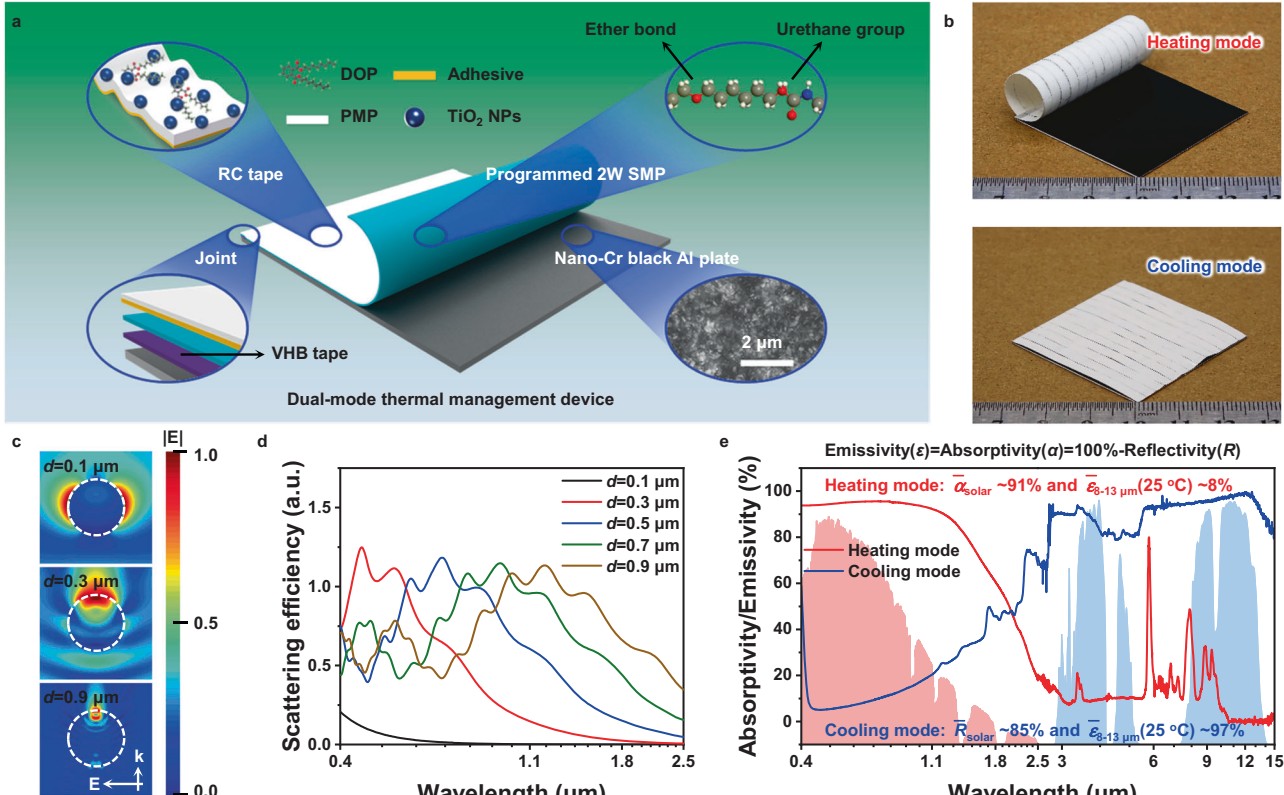

**Fig. 3 | Structure and spectral characteristics of high performance dual-mode thermal management device. a** Structural illustration of dual-mode thermal management device. Nano-Cr black Al plate is the solar collector with an electromagnetic spectrum close to ideal for solar heating. The functional layer for radiative cooling in RC tape is composed of DOP-modified PMP matrix and TiO$_2$ NPs fillers. The adhesive layer ensures the integration of the interface between the RC tape and temperature-sensitive actuator during complex and repeated deformation process. A piece of narrow VHB tape, used as the only joint part between solar heating and radiative cooling layers, reserves the maximum effective area for dual-mode thermal management. The inset of SEM image shows that nano-chromium oxide powders are uniformly distributed on the aluminum plate. **b** Optical images of dual-mode device in heating and cooling modes. **c** Cross-sectional view of a light field (magnitude of normalized electric field component of light) around a rutile TiO$_2$ sphere with different diameters ($d$). The wavelength of incident light is 475 nm, corresponding to the maximum energy density of solar radiation (ASTM G173). Electric field of the incident light and wave vector of the incident light are represented symbolically by E and k, respectively. **d** Simulated scattering cross-section spectra of TiO$_2$ spheres with different diameters in PMP matrix. **e** Absorptivity/Emissivity ($\alpha/\varepsilon$) of dual-mode thermal management device in heating and cooling modes, respectively.

energy input, but also lead to high efficiency in both heating and cooling modes for our dual-mode thermal management device.

## Design/selection of the materials/layers for heating/cooling mode

For dual-mode thermal management, an important aspect of the functional parts in the device is to achieve different electromagnetic spectrums with high selectivity required for solar heating and radiative cooling. Figure 3a illustrates the structure of a dual-mode thermal management device. Here, we introduced an aluminum plate coated with nano-chromium oxide powders (nano-Cr black Al plate) in the design of the dual-mode thermal management device for solar heating layer. The uniformly distributed nano-chromium oxide powders act as an absorbent and mirror agent to ensure high solar absorption and low infrared emission (inset in Fig. 3a). Due to plasmon resonances, the sunlight undergoes non-radiative damping in chromium oxide powders, and is further high-efficiency transformed into heat[12].

The radiative cooling in the dual-mode thermal management device is mainly achieved by home-made stretchable radiative cooling tape (RC tape) with excellent performance. The functional layer for radiative cooling in RC tape is made of dioctyl phthalate (DOP)-modified poly(4-methyl-1-pentene) (PMP) encapsulating rutile titanium dioxide nanoparticles (TiO$_2$ NPs) (Supplementary Fig. 1). PMP is an excellent solar transparent polymer with a wavelength-independent refractive index of 1.46 from visible to near-infrared range

(Supplementary Fig. 2), while the refractive index of rutile TiO$_2$ NPs is much higher (>2.39) than that of PMP[32]. The large difference of refractive index is a condition required for multiple scattering and internal reflection in the composite matrix. As corroborated by finite difference time domain (FDTD) simulation, the smaller TiO$_2$ NPs are more capable of redirecting incident light (Fig. 3c). On the other hand, the scattering center wavelength shows a red-shift trend with the increase in diameter of TiO$_2$ NPs (Fig. 3d). As scattering center with high refractive index, TiO$_2$ NPs with broad size distribution are able to produce the required scattering wavelength range covering the entire solar radiation, because of the collective effect of multiple Mie resonances (Fig. 3d and Supplementary Fig. 3). In addition, large amounts of infrared absorption peaks from various characteristics bonds in DOP-modified PMP, TiO$_2$ NPs, adhesive, and even shape memory polymer (materials for temperature-sensitive actuating layer), provide enough infrared radiation for transferring heat into outer space (Supplementary Fig. 4). The optimized RC tape can reflect >90% of solar radiation and have high absorptivity/emissivity of ~96% in the mid-infrared atmospheric window (8–13 μm) (Supplementary Fig. 4).

Nano-Cr black coated Al plate is black to absorb sunlight, and RC tape is glossy-white to reflect sunlight. Benefited from this, the device shows a drastic difference in visual appearance between heating and cooling modes (Fig. 3b). As shown in Fig. 3e, the device in heating mode can absorb ~91% of solar radiation and there is almost no mid-infrared emission (~8%) in the wavelength range of 8–13 μm. Such a

huge difference in the spectral characteristics of the device in the two modes lays the foundation for the zero-energy intelligent dual-mode thermal management device (Supplementary Fig. 5).

## Design of automatic actuation material/layer

To fully realize such an intelligent and automatic dual-mode thermal management device, there must be an auto-switching mechanism applied to the device. This is achieved with a temperature-triggered intelligent auto-switch using a temperature-sensitive layer with reversible shape memory sandwiched between the heating and cooling layers. The core mechanism of this actuation is to minimize the internal stress at the interface between the radiative cooling layer and the actuating layer, during reversible shape evolution of the actuating layer with temperature. Herein, two-way shape memory polymer (2 W SMP) is the key material for realization of temperature-triggered intelligent switch, which can be synthesized easily by a one-step esterification reaction of three monomers (polytetrahydrofuran (PTHF), polycaprolactone (PCL), and hexamethylene diisocyanate (HDI)) on a catalyst (dibutyltin dilaurate (DBTDL)) with almost 100% yield (Supplementary Fig. 6). The appearance of typical urethane group in the reaction product confirms successful synthesis of polyurethane prepolymer (Supplementary Fig. 7). The reaction product is then transferred directly to a stainless-steel petri dish to fully evaporate solvent at room temperature to get the required as-prepared 2 W SMP film for further preparation of the actuating layer later.

The temperature-triggered reversible shape memory performance is achieved after a programming process (Supplementary Fig. 8). During the heating-cooling cycles, there is a spontaneous and reversible length-shifting between shrinkage and elongation as expected, which is caused by the reversible melting-crystallization process of partial segments in the polymer (Fig. 4a). Remarkably, programmed 2 W SMP in stretching direction shrinks when heated and expands when cooled. A tight laminate could be formed by attaching a piece of same-sized RC tape to the programmed 2 W SMP at the shrinking state. Thanks to the huge difference of length along programming direction between RC tape and programmed 2 W SMP caused by the abnormal shrinkage behavior of the programmed 2 W SMP, the laminate could bend to RC tape side when cooled. As shown in Fig. 4b, the coiled laminate gradually unfolds until it is completely flat as the temperature increases. Notably, the bending angle starts to reduce slowly in the heating process. Once the temperature is higher than triggering temperature, the bending angle decreases sharply. This sharp angle change is determined by the melting of partial crystalline structure in programmed 2 W SMP (Fig. 4a). This ensures that RC tape-

2W SMP laminate keeps in coiled state at low temperature when heating is needed and unfolded state at high temperature when cooling is needed without excessive bending to programmed 2 W SMP side, achieving the designed automatic and temperature-triggered switching. A hysteresis of bending angle exists during a heating-cooling cycle, which is from the difference between melting temperature and crystallization temperature of programmed 2 W SMP. The triggering temperature of RC tape-2W SMP laminate could be adjusted by the molecular weight ($M_w$) of PCL monomers, according to the requirements of practical scenario (Supplementary Figs. 9 and 10). For PCL with $M_w = 10{,}000$, the triggering temperature is in the range of 23–24 °C, around the comfortable temperature zone for human living (Fig. 4b). In addition, RC tape-2W SMP laminate exhibits excellent cyclability during repeated heating-cooling process, indicating good stability in long-term operation (Supplementary Movie 1 and Supplementary Fig. 11).

Then, several RC tape-2W SMP laminates are placed side by side and bonded together to form a large-sized film exactly covering the nano-Cr black Al plate (Supplementary Note 2 and Supplementary Fig. 21). A piece of narrow VHB tape, as the only joint part between solar heating and radiative cooling layers, reserves the maximum effective area for dual-mode thermal management. We also demonstrated the robustness of dual-mode device repeatedly switching between heating and cooling modes (Fig. 4c). Briefly, the reversible shape transformation of RC tape-2W SMP laminate realizes the maximum percentage of active area in dual-mode device, which is conductive to achieving the best effect of thermal management in heating and cooling modes, respectively. And the temperature-sensitive trigger mechanism makes it possible for dual-mode devices to intelligently and freely switch between two thermal management modes without any external energy consumption (Supplementary Movie 2).

## Thermal management performance of dual-mode device

To estimate working efficiency of this dual-mode device for both heating and cooling modes, a Joule heating-based measurement system is designed to monitor heat flux (Supplementary Fig. 18). The Peltier device combined with a fan is used as a stable cold source in the system. An indoor experiment has been carried out first with the solar simulator (AM 1.5) before field test outdoors. The solar heating power and radiative cooling power of the dual-mode device is tested for five cycles (Supplementary Fig. 19c). For solar heating, the average heat flux of dual-mode device achieves $933.6 \pm 13.7\,\mathrm{W\,m^{-2}}$, which is almost consistent with the theoretical value of dual-mode device in heating mode, approximately equal to 94% of solar radiation (ASTM G173)

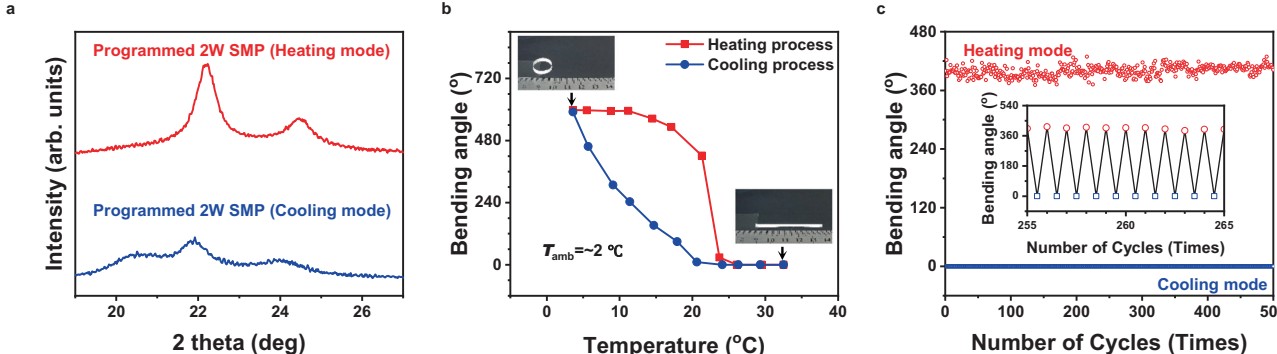

**Fig. 4 | Reversible shape memory performance. a** X-ray diffraction spectrums of programmed 2 W SMP in heating and cooling modes, respectively. **b** Bending deformation performance of RC tape-2W SMP laminate as a function of temperature of heating plate. The molecular weight of PCL monomer is 10,000. The inset of optical images shows that RC tape-2W SMP is in coiled state at low temperature and unfolded state at high temperature. **c** Reversible bending deformation of RC tape-2W SMP film array in the dual-mode device as a function of the number of cycles between heating and cooling modes.

(Fig. 3e). Similarly, for radiative cooling, the average heat flux is $-94.4 \pm 42.8$ W m$^{-2}$, which is ~55% of theoretical value in cooling mode. Both a certain difference and fluctuation may be from insufficient thermal contact between RC tape-2W SMP film and nano-Cr black Al plate. More details about theoretical model calculation are described in Supplementary Note 3. Furthermore, the dual-mode device spontaneously switches between heating and cooling modes by responding to the temperature, without external energy consumption. After repeated switching, whether it is in heating mode or cooling mode, the thermal management performance has no obvious degradation.

Further, we performed a daily field experiment in the real-world scenarios outdoors to test its truly practical thermal management performance in a real environment (located on the campus of Nankai University in Tianjin (38.99 N, 117.34E), China) (Supplementary Note 5). Two same systems are set in parallel for comparison (Supplementary Fig. 18). One copper (Cu) plate is covered by our dual-mode device, and the other is covered by a same-sized aluminum (Al) foil as a control group, because its solar absorption and infrared emission are close to zero (Supplementary Fig. 13). The heater in the system for dual-mode device is connected to a constant current source, and the other is connected to a feed-back control program to maintain the temperature of Al foil the same as that of the dual-mode device (Supplementary Fig. 23). Shown in Fig. 5a are the three heat flux curves recorded for the solar radiation, dual-mode device in heating and cooling modes, respectively. The solar heating power continues to increase and achieves close to 958.7 W m$^{-2}$ with stronger and stronger solar radiation, where the real-time solar-thermal conversion efficiency always remains around 91%. In addition, the average radiative cooling power around noon reaches 126.0 W m$^{-2}$ under the normal-incidence solar radiation >850 W m$^{-2}$. Considering the reduced ambient thermal radiation and the inevitable heat convection and conduction (Supplementary Note 4), the measuring heating flux data of dual-mode devices in both heating and cooling modes outdoors matches well with the indoor experimental results. These results demonstrate that our dual-mode device could achieve rather high-efficiency thermal management performance repeatedly in both solar heating and radiative cooling modes, and automatically switch between them according to the temperature. During the whole process, including working and switching, zero external energy is required. The dual-mode device is feasible to work in the real world throughout different seasons of the entire year. As far as we know, the design of this dual-mode thermal management device with these features combined together, including two thermal management modes, zero-energy consumption, and intelligent and free switching, has not been reported in the literature (Supplementary Table 1).

Referred to historically meteorological data, we calculated monthly produced heat and cold of dual-mode device in heating and cooling modes, respectively, to quantitatively predict the potential impact of dual-mode device on energy saving (Supplementary Note 6). With the periodic change of relative position between the earth and the sun, the heating and cooling capacities of the dual-mode device in different months show a certain regularity. Taking Tianjin, a typical continental-monsoon-climate city, as an example, the total solar radiation and average temperature increase first and then decrease together in 1 year (Fig. 5b and Supplementary Table 2). Even in colder winter, the dual-mode device is still able to produce considerable heat (>0.15 GJ m$^{-2}$), thanks to its high solar-thermal conversion efficiency, although the total solar radiation is very low. The cooling capacity is mainly determined by temperature, less affected by the solar radiation. The peak reaches 0.24 GJ m$^{-2}$ in July and August, just corresponding to the hot summer. The year-round accumulated energy saving exceeds 2.9 GJ m$^{-2}$ in prediction. The maximum energy saving for heating in January will happen at $\alpha_{solar} = 100\%$ and $\varepsilon_{infrared} = 0\%$, and that for cooling in July occurs at $\alpha_{solar} = 0\%$ and $\varepsilon_{infrared} = 100\%$ (Fig. 5c, d). It agrees well with our proposed two ideal high-selectivity

electromagnetic spectrums (Fig. 1b). Compared with temperature-responding thermal management devices (including windows and coatings) reported in the literature[30,33–38], our dual-mode device could reach 91% of solar absorptivity and 8% of infrared emissivity for heating, and 90% of solar reflectivity and 97% of infrared emissivity for cooling, which is very close to the ideal electromagnetic spectrums. This great improvement of spectral selectivity puts our device in a different operational space and sets a new mark for dual-mode radiative thermal management. Some cities are selected to represent typical terrestrial climatic zones around the world (Supplementary Fig. 25 and Supplementary Table 3). It can be seen that the dual-mode device has significant effects of energy saving in almost all climate zones, whether in heating mode or cooling mode. We assumed that the dividing temperature between heating and cooling modes is 17 °C, which is approximately equal to the average temperature of Beijing in spring and autumn. The corresponding energy-saving map is shown in Fig. 5e. In January, the weather is cold in most areas north of the Tropic of Cancer, and the dual-mode device works in heating mode. In general, the closer to the Tropic of Cancer, the more energy for heating can be saved from solar-thermal conversion of dual-mode device. It is consistent with the change of solar radiation as a function of the latitude. In contrast, the weather, in most areas located in the south of the Tropic of Cancer, is warm or even hot in January. Dual-mode device in cooling mode achieves good effect of energy saving for cooling, especially in the area near the Tropic of Capricorn, where it is in summer. The above analysis describes the great potential of the dual-mode device in terms of global thermal management and energy saving.

A real-time demonstration of the high-performance temperature control by the dual-mode device outdoors is shown in Fig. 5f. With alternative applying and removing of a constant Joule heating power, the dual-mode device spontaneously switches between cooling mode and heating mode by perceiving temperature (Supplementary Fig. 27). A bare Cu plate with an almost invariable electromagnetic spectrum is used as a control group. As expected, the Cu plate covered by the dual-mode device in heating mode is obviously ~6 K higher than the bare one under the solar radiation, when it is cool. And when it is hot, a temperature reduction close to 15 K is realized by the dual-mode device in cooling mode. Even at dark night, the dual-mode device could also preserve heat due to the low infrared emission in heating mode, and still efficiently produces cooling in cooling mode (Supplementary Fig. 28). A total of ~21 K reduction of temperature fluctuation strongly and visually shows the ability to control temperature for the dual-mode device.

## Discussion

In summary, we reported an intelligent and zero-energy dual-mode radiative thermal management device with two sets of spectral characteristics close to the ideal spectrums for solar heating and radiative cooling, which is able to automatically switch to the right mode depending on the ambient temperature. In the real world, the device can achieve an average heating power of ~859.8 W m$^{-2}$ (corresponding to average solar-thermal conversion efficiency of ~91%) in cold and an average cooling power of ~126.0 W m$^{-2}$ in hot, because it has two different high-selectivity spectral characteristics. Owing to the temperature-triggered reversible auto-switch, the device could intelligently choose an appropriate mode to get the best temperature control results. This design of dual-mode device maximums the zero-energy advantage of solar heating and radiative cooling in thermal management, which has not been reported in the literature as far as we know.

As a zero-energy design, the dual-mode thermal management device takes full advantage of renewable energy in nature, solar heat and space cold, thereby very well-suitable for open areas, such as roofs of large-scale buildings. The generated thermal

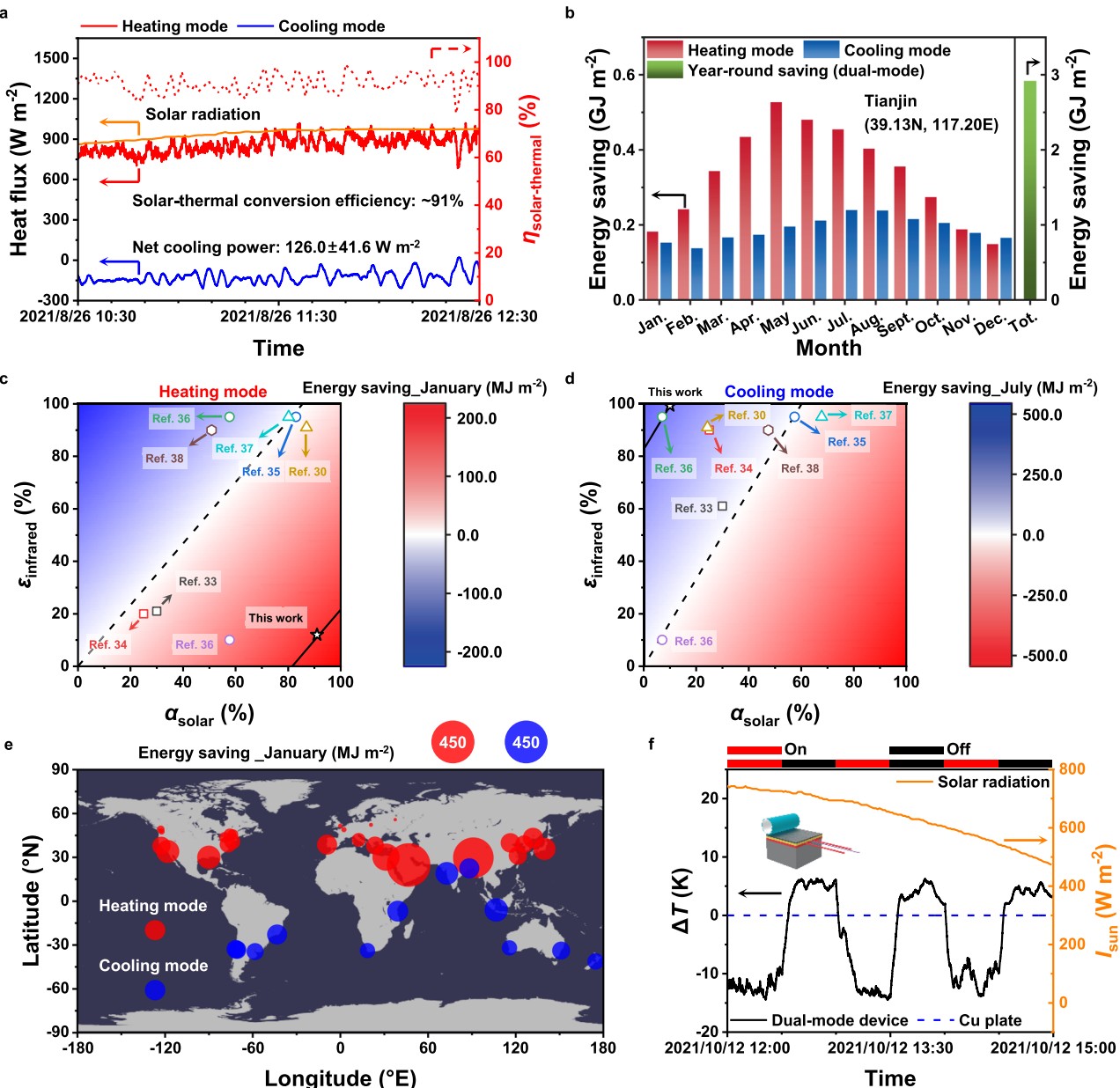

**Fig. 5 | Thermal management performance of dual-mode device. a** Continuous time-resolved solar heating power (red line) and radiative cooling power (blue line) measured in field test. The solar-thermal conversion efficiency ($\eta_{\text{solar-thermal}}$) fluctuates around ~91% (red dash line) according to real-time solar radiation (orange line). **b** Modeled monthly all energy saving of dual-mode device in heating (red) and cooling (blue) modes in Tianjin for 1 year and year-round energy saving (green). The critical temperature for dividing heating and cooling modes is assumed as 17 °C, which is approximately equal to the average temperature of Beijing in spring and autumn. Heating mode: January–April, October–December. Cooling mode: May–September. **c, d** The effects of solar absorptivity ($\alpha_{\text{solar}}$) and infrared emissivity ($\varepsilon_{\text{infrared}}$) on (**c**) heating energy saving in January and (**d**) cooling energy saving in July in Tianjin. Solar absorptivity with the corresponding infrared emissivity of the dual-mode device (star) is compared with those of temperature-responding device (doped-vanadium dioxide (VO$_2$), square; hydrogel, circle; phase-changing polymer, triangle; other materials, hexagon) in the literature. **e** Modeled energy-saving (radius of circle) map for some cities with dual-mode device in heating mode (red circle) or cooling mode (blue circle) in January. **f** Real-time temperature difference ($\Delta T = T_{\text{sample}} - T_{\text{Cu plate}}$) of dual-mode device ($T_{\text{sample}}$, black line) compared with 200-µm-thick Cu plate ($T_{\text{Cu plate}}$, blue dash line) under solar radiation ($I_{\text{sun}}$, orange line). As Joule heating power is repeated to be on-off, the dual-mode device switches between cooling mode and heating mode by perceiving temperature.

energy could be used as a direct source for space temperature control by arranging a large number of devices on the roof, but this is a low-efficiency selection and just able to regulate temperature where the space is close to the roof. Referring to the reported studies, the application of radiative thermal management to initial temperature control of heat-transfer fluid in active thermal management systems will significantly improve efficiency in thermal energy use. In general, this idea represents a system-level

approach to renewable energy generation and efficiency in the future. As far as an individual device is concerned, how to further improve its thermal management performance and weather resistance in the real scenario is the top priority. However, such a real zero-energy dual-mode device would have great and practical potential for global thermal management and energy saving, and provides a renewable zero-energy platform to realize the goal of Net Zero Carbon 2050.

## Methods

### Preparation of radiative cooling tape

Radiative cooling tape (RC tape) was prepared by multiple blade coating. First, 0.5 g of poly-4-methyl-1-pentene (PMP) particles (Mitsui Chemicals, MX002) were dissolved in 20 mL of cyclohexane solvent (Aladdin, AR 99.5%) by stirring at 60 °C. Then, 1.355 g of rutile titanium dioxide nanoparticles ($TiO_2$ NPs) (Shanghai Yaoyi alloy material Co. Ltd) and 0.148 g of dioctyl phthalate (DOP) (Aladdin, AR 99.0%) were mixed with PMP solution in proportion by tip ultrasonication (500 W, 30 min) to make a precursor solution. The PMP-DOP-$TiO_2$ solution was blade-coated onto a clean stainless-steel substrate to fabricate the uniform liquid film between two transparent tape spacers, which is placed on an 80 °C heating plate immediately to rapidly evaporate solvent. The thickness of the RC tape (75 μm) was determined by the repeated numbers of blade coating-drying process. Water soluble glue (Wen Ding adhesive Co. Ltd, #803) was blade-coated onto the as-prepared composite film to form the adhesive layer. After heating to 80 °C to remove the residual solvent, RC tape was easily removed from the substrate. The RC tape film can be tailored to desired shape and further fabricated into diverse products.

### Synthesis of two-way shape memory polymer

Two-way shape memory polymer (2 W SMP) was synthesized by esterification reactions between monomers on catalyst (Supplementary Fig. 6). First, polytetrahydrofuran (PTHF, average $M_w = 2900$, Sigma-Aldrich) and polycaprolactone (PCL, average $M_w = 10,000/36,000$, Aladdin) (or polycaprolactone-diol (PCL-diol, average $M_w = 2000$, Aladdin)) were fully dissolved in trichloromethane ($CHCl_3$, AR, Tianjin Bohua Chemical Reagents Co., Ltd) in proportion by stirring at room temperature. Then, hexamethylene diisocyanate (HDI, 99%, Aladdin) and dibutyltin dilaurate (DBTDL, 95%, Aladdin) were added into the solution in sequence, which was stirred constantly for 3.5 h at room temperature. The molar ratio of three monomers ($n_{PTHF}$:$n_{PCL}$:$n_{HDI}$) was 9:1:20. The amount of DBTDL catalyst was 1% of the total weight of three monomers. During the process, monomers polymerized gradually to form a 2 W SMP. The product for the esterification reaction was poured into a horizontal stainless-steel petri dish. After completely volatilization of the solvent at room temperature, a 2 W SMP film existed at the bottom of the petri dish, which can be further cut into any shape as required.

### Fabrication of dual-mode thermal management device

2 W SMP synthesized by PCL-like monomer with molecular weight of 36,000 was chosen as an example. An as-prepared 2 W SMP strip was stretched out to five times its length at 90 °C (programming temperature) and locked until cooling to room temperature (low temperature). Then, the stretched 2 W SMP strip shrunk to a certain degree along stretching direction at 55 °C (high temperature) to finish programming treatment. A piece of same-sized RC tape was attached to the programmed 2 W SMP strip at 55 °C. This RC tape-2W SMP laminate was coiled at room temperature and unfolded at 55 °C. Several RC tape-2W SMP laminates were placed side by side at 55 °C, and boned together by some pieces of narrow transparent tape to form a significantly sized film. This film was fixed on a same-size aluminum plate coated with nano-chromium oxide powders (nano-Cr black Al plate, KNEAR) by a piece of narrow VHB tape to prepare a dual-mode device (Fig. 3a).

### Characterization

The reflectance (R) of the dual-mode device in different modes was measured using an ultra-violet-visible-near-infrared (UV-NIR) spectro-photometer (Agilent, Cary 5000) with an integrating sphere and a Fourier transform infrared (FT-IR) spectrometer (Perkin Elmer, Frontier Optica) with an integrating sphere [PIKE, MCT Mid-IR Integrated sphere]. The absorptance/emittance ($\alpha/\varepsilon$) was calculated using 100%-R (0% transmissivity determined by Al plate). The surface morphology was observed using a scanning electron microscope (JEOL, JSM-7800). X-ray diffraction data was achieved by X-ray powder diffraction instrument (Rigaku Smart Lab SE).

### Measurement of heat flux (indoor)

Supplementary Fig. 18 shows the apparatus to quantitatively estimate the ability to thermal management including solar heating and radiative cooling. From top to bottom, it involves a dual-mode device, a copper (Cu) plate (length: 40 mm, width: 40 mm, thickness: 0.2 mm), a heater, a Peltier device, and a fan. Double-side tape is applied to ensure good mechanical stability among the dual-mode device, the Cu plate, the heater, the Peltier device, and the fan. An aluminum (Al) foil-coated PET with a square opening of 40 mm × 40 mm in center is attached to the fan to avoid degradation of ability to heat dissipation under solar radiation. The indoor experiment was done with the solar simulator (AM 1.5). During the whole experiment, the working Peltier device combined with the fan was used as a stable cold source in the apparatus. Under simulated sunlight, a constant Joule heating power was turned on and off alternately, resulting in the auto-switch between cooling mode and heating mode for dual-mode device. Then, in the dark, the heater connected with a proportion-integration-differentiation (PID) program controlled the dual-mode device at the same temperature as that under simulated sunlight, including heating and cooling modes. The steady-state temperature of dual-mode device in the dark matched well with that under simulated sunlight, no matter heating and cooling modes (Supplementary Fig. 19a). The difference of Joule heating power between two scenes was the corresponding solar heating power (positive heat flux) and radiative cooling power (negative heat flux) under simulated sunlight (Supplementary Fig. 19b). The ambient temperature was relatively steady during the whole experimental process. To estimate thermal management performance of the device in two modes, we chose 2 W SMP with higher transition temperature (synthesized by PCL-like monomer with molecular weight of 36,000, rather than 10,000).

### Measurement of heat flux (outdoor)

The apparatus for measuring solar heating power (positive heat flux) and radiative cooling power (negative heat flux) is the same as the one used in simulated scenes. To record heat flux data in real time, the same two apparatus were set in parallel. One Cu plate was covered by a dual-mode device, the other was covered by a same-sized Al foil, as a control group, due to its strong reflection for solar radiation and infrared radiation (Supplementary Fig. 13). The heater in the apparatus for dual-mode device was connected with a constant current source, and the other was connected with a feedback-controlled heating system to always maintain the temperature of Al foil at the same temperature as the dual-mode device. Because Al foil has a little solar absorption (the weighted-average solar absorptivity of ~6.5% estimated from simulated spectrum (Supplementary Fig. 13)), the heat flux ($\Phi_q$) should be calibrated based on this additional solar-thermal conversion. Solar heating power (positive heat flux) or radiative cooling power (negative heat flux) is calculated by $\Phi_q = P_{Al} - P_{device} + \alpha_{Al}I_{sun}$, where $P_{Al}$ is Joule heating power per area applied on Al foil, $P_{device}$ is Joule heating power per area applied on device, $\alpha_{Al}$ is weighted-average solar absorptivity of Al foil and $I_{sun}$ is the intensity of solar radiation. For solar heating, Peltier devices and fans combined with them operated under the same conditions to maintain a lower temperature, which endows the dual-mode device in heating mode. For radiative cooling, Peltier devices and fans were still in the working state. By the heater, an appropriate constant Joule heating power was applied on the dual-mode device to make the device switch to cooling mode. A weather station was placed near the apparatus to record the weather conditions at the testing position. Because the experiment was done in summer and the ambient temperature was relatively

steady during the whole experimental process, we chose 2 W SMP with higher transition temperature (synthesized by PCL-like monomer with molecular weight of 36,000, rather than 10,000) to show dual-mode thermal management performance.

## Data availability

The data that support the findings of this study are available from the corresponding author upon reasonable request.

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

## Acknowledgements

This work is supported by the National Key R&D Program of China (2020YFA0711500 (R. M.) and 2020YFA0711501 (R. M.)), the National Natural Science Foundation of China (51973095 (R. M.) and 52011540401 (R. M.)).

## Author contributions

Q.Z., R.M. and Y.C. conceived and designed the experiments; Q.Z. prepared RC tape and carried out the simulation of FDTD; Q.Z., Y.L. and Y.W. synthesized 2 W SMP and prepared RC tape-2W SMP film; Q.Z., Y.L., Y.W. and S.Y. fabricated the experimental setup and performed

the measurements; Q.Z., Y.L., Y.W., S.Y., C.L., R.M. and Y.C. analyzed and interpreted the data; The manuscript was mainly prepared by Q.Z., Y.L., R.M. and Y.C. and all authors participated in the manuscript preparation and comment on the manuscript. R.M. and Y.C. supervised the work.

## Competing interests

The authors declare no competing interests.
