## [Peer Review File · Nature Communications]

Temperature-dependent dual-mode thermal management device with Net Zero energy for year-round energy savingREVIEWER COMMENTS

Reviewer #1 (Remarks to the Author):

This work demonstrates an energy-free dual-mode thermal regulation device. This device is composed of one trilayer actuator and one solar energy absorbing layer. The trilayer actuator consists of a layer of radiative cooling (RC) tape with TiO₂ nanoparticles (NPs), an adhesive layer and a layer of two-way shape memory polymer (2W SMP). The cooling mode relies on the top surface of trilayer structure, in which the RC tape provides great IR emissivity while the TiO₂ NPs decrease the solar absorptivity. The heating mode is realized using the solar energy absorbing layer with a low thermal emittance, which is an Al plate coated with nano chromium oxide powders. The energy-free transition between those two modes is achieved using the temperature-sensitive 2W SMP. Though the application of this technology seems to be limited due to its mechanical actuation of a fragile film, its performance is excellent and the whole work is well-organized. I would like to encourage the authors to discuss the following issues to further improve this manuscript.

1. Perhaps the most fundamental problem for this work is the application. The soft trilayer structure might be fragile to wind/rain/snow/etc. In this work, the outdoor experiment was conducted with a wind speed of no more than 2 m/s.
 - a) How to protect this device from heavy rains? Will the surface tension of water prevent the trilayer actuator from opening?
 - b) Is there any possible approach to prevent strong winds from damaging this device?
2. The small heat conduction between the trilayer structure and the nano-Cr back Al plate limits its cooling performance. How to improve it?
3. What is the thermal conductance of the trilayer structure? I am curious about the temperature gradient in the trilayer structure in real applications. For example, if the environment is rather cold but this device is attached to a hot surface, it is possible that the small thermal conductance of the trilayer leads to a cold trilayer structure so that it is (partially) opened. A similar problem will happen if you want to heat up a cold surface on a hot summer day, where the trilayer cannot be efficiently cooled down by the substrate.
4. Furthermore, what is the heating/cooling power of this device in a "semi-open" state between 5 °C and 24 °C in Fig. 3b?
5. On page 9, the authors wrote "the scattering center wavelength shows a blue-shift trend with the increase in diameter of TiO₂ NPs". Is it blue-shift or red-shift?
6. On page 10, the authors claimed that "The optimized RC tape can reflect more than 90% of solar radiation...", but why this reflection becomes 85% in Fig. 2e?
7. I feel confused by Fig. 4f as well as the supplementary Fig. 22b, where the heating and cooling modes (denoted by the color of the curve) seem to be dependent on the ON/OFF of Joule heating rather than the actual temperatures? Whenever the authors switched ON or OFF the Joule heating power, the definition of heating/cooling mode changes "instantly" without any delay? I assume that a temperature change should occur first until reaching the "transition temperature", and then the device working mode changes accordingly. Time-dependent temperatures of both this device and the copper plate are necessary to avoid confusion.
8. What is the thickness of DOP-modified PMP layer in supplementary Fig. 3a and 3b?

Reviewer #2 (Remarks to the Author):

In this paper, a dual-mode radiation cooling / absorption heating structure is designed based on the deformation maladjusted bilayer film. The design idea of the structure is novel and can overcome the dilemma of radiation cooling materials in the low-temperature environment. However, some details of the experimental design in this paper need to be explained again:

1. The outdoor test takes copper sheet as a reference, which is very different from the actual situation. The copper sheet will absorb heat and heat up under the irradiation of sunlight, but the atmospheric temperature does not reach the same temperature at this time. In this case, the heat absorption power and radiation power of the dual-mode device are unreasonable, which will lead to underestimate the heat absorption power and overestimate the radiation power.
2. When radiation cooling materials can save energy consumption at high temperature, they need to be able to cool below the ambient temperature. Materials have radiation ability does not mean that they can cool below the ambient temperature. The comparison between the outdoor test temperature

of the radiation film and the ambient temperature is not given in this paper, so it is impossible to judge whether it has the ability to reduce the temperature below the ambient temperature.

3. Aluminum plate and copper plate are selected as references for indoor test and outdoor test respectively, because they have enough low thermal radiation capacity. However, if there is oxidation on the surface of aluminum and copper plates, their radiation ability are often obvious. The infrared radiation spectrum of them is not given in this paper. In addition, why should we choose different materials for comparison indoors and outdoors?

4. The heat absorption power and radiation power of dual-mode device are evaluated by the input power of heater and thermoelectric device. Does it not need to consider the working efficiency of these devices? Especially the efficiency of thermoelectric refrigeration.

5. Theoretical calculations are based on Formula 1, but it is not clear how to use some key parameters. For example, when calculating indoor heat absorption and radiation power, what is the used environmental radiation spectrum?

Reviewer #3 (Remarks to the Author):

The authors developed an innovative way to achieve an auto-switched and zero-energy dual-mode radiative thermal management device and demonstrated the function through folding and unfolding the laminate structure to the variation of ambient temperature. The idea is interesting but several questions should be addressed before making decision.

1. The results in this work only show the temperature reduction close to 15 K in cooling mode and growth close to 6 K in the heating mode by just comparing with a bare Cu plate. Therefore, the description in abstract 'A practical demonstration shows that the temperature fluctuation is reduced by ~21 K.' should be specified by comparing with a bare Cu plate. However, photonic devices designed for thermal management are generally used in outdoor structures, such as buildings, cars, and space aerocrafts, as well as soft and wearable textiles, etc. Hence, I think the authors should give a little more comparison of the thermal performance with, for instance, two or three other materials.

2. The authors stated that the average solar heating power and radiative cooling power are ~859.8 W/m² and ~126.0 W/m² in heating and cooling modes, respectively, which correspond to folding and unfolding of the laminate. My question is, is the result from all of the area in unfolding state? While in folding laminate, especially in the case of film array, both in horizontal and vertical direction, only a part of the area will contribute to the heating effect because the other part of the area will be covered by the coiled laminate. As such, the values of power of thermal load of heating mode should be revised.

3. Since the present intelligent dual-mode devices are the folding and unfolding laminates, I think the size of a single plate will affect the effectivity of thermal loading, switching, and potential applications of the devices. Therefore, additional results from the devices with different sizes maybe more interesting.

4. The effect of incident angle of solar radiation should be discussed, especially in the cases that the size of laminates is not much smaller, because it will greatly affect the absorption.

5. Additionally, the authors also reported the other triggered temperature 'This RC tape-2W SMP laminate was coiled at room temperature and unfolded at 55 °C.' In fact, such triggered temperature is somewhat a little higher value in most cases. Can the devices also work as the ambient temperature is around 45 °C? 50 °C ?

6. Can the devices also work well as the substrate becomes to, for instance, soft or wearable textiles?

Reviewer #4 (Remarks to the Author):

The paper introduces a high-performance, intelligent auto-switched and zero energy dual-mode radiative thermal management device. It describes in details the novel features of the design. It demonstrates its performance via modelling, lab tests and in real-file outdoor settings.

The provided evidences support the conclusions and the novel technology sounds interesting and with potential.

The authors build the argumentation behind the novel technology on

“the fact of the global total primary energy demand is close to a 15-billion-tonne oil equivalent in 2019, and nearly 50% of energy consumption is merely used for daily heating and cooling (...) Therefore, it is particularly important and imperative to develop various feasible high-performance thermal management technologies with low or even zero energy consumption, which is able to reduce fossil energy demand and further emission of greenhouse gases”

and presents its heating and cooling potential for various locations. This is very right. Yet, the conclusions lack a section describing the potential application of this new device. It would significantly enhance the value of the paper if suggestions, for how and where this technology can be applied, were part of the paper.

I hope these suggestions will improve your manuscript and make it even more valuable to the field.

Dear Reviewers,

We greatly appreciate the thoughtful comments of the reviewers. Herewith we are submitting the revised manuscript entitled, “Temperature-Dependent Dual-Mode Thermal Management Device with Net Zero Energy for Year-Round Energy Saving” for publication as an Article in *Nature Communications*.

The list of changes starts on the next page. Additional experimental data and analysis have been added to address the comments of reviewers. For clarity, the reviewers’ comments are written in *italics* below, and our responses to the reviewers are in **dark red** while the modifications to the manuscript and SI are in **blue**.

Thank you very much for your time.

Sincerely,

Professor xxx

Reviewer Comments

Reviewer #1 (Remarks to the Author):

This work demonstrates an energy-free dual-mode thermal regulation device. This device is composed of one trilayer actuator and one solar energy absorbing layer. The trilayer actuator consists of a layer of radiative cooling (RC) tape with TiO₂ nanoparticles (NPs), an adhesive layer and a layer of two-way shape memory polymer (2W SMP). The cooling mode relies on the top surface of trilayer structure, in which the RC tape provides great IR emissivity while the TiO₂ NPs decrease the solar absorptivity. The heating mode is realized using the solar energy absorbing layer with a low thermal emittance, which is an Al plate coated with nano chromium oxide powders. The energy-free transition between those two modes is achieved using the temperature-sensitive 2W SMP. Though the application of this technology seems to be limited due to its mechanical actuation of a fragile film, its performance is excellent and the whole work is well-organized. I would like to encourage the authors to discuss the following issues to further improve this manuscript.

1. Perhaps the most fundamental problem for this work is the application. The soft trilayer structure might be fragile to wind/rain/snow/etc. In this work, the outdoor experiment was conducted with a wind speed of no more than 2 m/s.

a) How to protect this device from heavy rains? Will the surface tension of water prevent the trilayer actuator from opening?

b) Is there any possible approach to prevent strong winds from damaging this device?

We thank the reviewer for this insightful comment. In the additional experiments, the recorded maximum wind speed is over 4 m/s (Supplementary Fig. S24 in response to Comment #4). Strong wind will produce obvious fluctuations in measuring values of heat flux. Therefore, we did field tests on days with clear skies and calm winds.

Considering that the filling of TiO₂ NPs would reduce the stretchability and flexibility of pure PMP film, a moderate amount of DOP was introduced into PMP encapsulating TiO₂ NPs. The stress-strain curve shows that the introduction of DOP could significantly improve the mechanical properties of TiO₂/PMP composite film (Supplementary Fig. 1). As a result, DOP-modified PMP encapsulating TiO₂ NPs is flexible enough to undergo the impact from the norm wind or rain.

The damage to the device from heavy rains could be divided into two categories: physical impact and chemical corrosion. The flexibility protects coiled RC tape-2W SMP film from continuous impact of numerous raindrops. We also make efforts to further reduce radius of curvature of coiled RC tape-2W SMP film by increasing the strain of programmed 2W SMP. On the one hand, this reduces the chance of collision between the film and raindrops. On the other hand, it could increase the effective area of the device in heating mode. In addition, RC tape (DOP-modified PMP encapsulating TiO₂ NPs), programmed 2W SMP and nano-Cr black Al plate are insoluble in water. It reduces the risk of breakage caused by raindrop corrosion.

The morphological evolution of RC tape-2W SMP film depends on its temperature. So, the film could unfold well when the temperature is higher than transition temperature of programmed 2W SMP, although it is hydrophobic. Even so, water between folded RC tape-2W SMP may increase the interfacial thermal resistance between the film and nano-Cr black Al plate. It will limit cooling performance of the device.

The tearing from drastic air flowing is the main reason for damage to the device on a windy day, especially for RC tape-2W SMP film. Similar to the protection in heavy rains, small radius of curvature is helpful by reducing the contacting area between coiled RC tape-2W SMP film and air. Moreover, we also consider replacing organic-inorganic composite in RC tape with flexible organic RC materials. The stronger resilience makes the device better resist the impact from strong winds.

Herein, we proposed and experimentally demonstrated a concept of zero-energy dual-mode radiative thermal management by trading morphological evolution for switching of high-selective spectral characteristics. However, there are still a lot of technological issues to be solved for future applications in various scenarios.

Supplementary Fig. 1 and the related description have been added in the revised manuscript and SI.

(Manuscript) Page 9, Line 204-206:

The functional layer for radiative cooling in RC tape is made of dioctyl phthalate (DOP)-modified poly(4-methyl-1-pentene) (PMP) encapsulating rutile titanium dioxide nanoparticles (TiO₂ NPs) (Supplementary Fig. 1).

(SI) Page 16, Line 315-323:

Supplementary Fig. 1 | Mechanical properties and hydrophobicity of DOP-modified PMP encapsulating TiO₂ NPs. a, Stress-strain curve for DOP-modified PMP/TiO₂ NPs composite

film with different volume fractions of DOP. The volume fraction of TiO₂ NPs is fixed at 50%. **b**, Stress-strain curve for DOP-modified PMP/TiO₂ NPs composite film with different volume fractions of TiO₂ NPs. The volume ratio between PMP and DOP is fixed at 4:1. **c-e**, Photographs of a RC tape showing its appearance (**c**) before, (**d**) during and (**e**) after wetting with deionized water. No changes are observed, indicating that RC tape is hydrophobic.

2. *The small heat conduction between the trilayer structure and the nano-Cr black Al plate limits its cooling performance. How to improve it?*

Thank you for this consideration. Inadequate thermal contact between RC tape-2W SMP film and nano-Cr black Al plate does limit the cooling performance of dual-mode device in cooling mode. Interfacial thermal resistance is so high that the generated cooling energy would not be transported and fully utilized by the underlying object.

Electrostatic force has been proven to be able to effectively enhance interfacial thermal conduction. Hsu et al. (*Nat. Commun.* **11**, 6101 (2020)) reported a dual-mode thermal management device with electrostatic force-controlled interfacial thermal resistance. As the macroscopic area and microscopic contact area both increase under Maxwell pressure, the improvement of thermal contact could boost the thermal management performance approaching the theoretical prediction: “The average thermal contact conductance can reach 9.5×10^2 W/(m²·K) at 2 kV applied voltage, and the corresponding temperature difference is suppressed to about 0.4 °C”.

Besides electrostatic interaction, magnetostatic force is another feasible way to effectively enhance interfacial thermal conduction. For example, we could construct a magnetic-nanoparticle layer attached to the bottom of 2W SMP, and use ferromagnetic material, such as nano-Cr black Fe plate, instead of nano-Cr black Al plate. In doing so, there is an improved thermal contact between the radiative cooling layer and the solar heating layer, when the device works in cooling mode.

The above discussion is only our preliminary assumption. However, this comment motivates us to do more investigations in a future study to optimize the structure for improving thermal contact between different functional layers in dual-mode device, thus maximizing its thermal management performance as predicted by theory.

3. *What is the thermal conductance of the trilayer structure? I am curious about the temperature gradient in the trilayer structure in real applications. For example, if the environment is rather cold but this device is attached to a hot surface, it is possible that the small thermal conductance of the trilayer leads to a cold trilayer structure so that it is (partially) opened. A similar problem will happen if you want to heat up a cold surface on a hot summer day, where the trilayer cannot be efficiently cooled down by the substrate.*

Thank you for this comment. The thermal conductivity of typical materials in RC tape-2W SMP laminate are referred to the data in the reported studies.

Polyurethane (2W SMP), 0.177 W/(m·K) (*Compos. Pt. B-Eng.* **42**, 2111–2116 (2011)).

Acrylic ester (adhesive), 0.21 W/(m·K) (*J. Appl. Polym. Sci.* e52629 (2022)).

PMP (matrix in RC tape), 0.17 W/(m·K) (*Polymer* **238**, 124423 (2022)).

Rutile TiO₂ (fillers in RC tape), 4.059 W/(m·K) (*J. Mol. Liq.* **302**, 112499 (2020)).

The estimated thermal conductivity of DOP-modified PMP encapsulating TiO₂ NPs is from 0.17 W/(m·K) (0 vol% of TiO₂ NPs) to 2.76 W/(m·K) (50 vol% of TiO₂ NPs), according to Maxwell-Eucken model (*Int. J. Heat Mass Transf.* **137**, 184–191 (2019)).

The morphology evolution of RC tape-2W SMP film is a continuous process. The morphology depends on its own temperature, rather than just the temperature of the covered object or the ambient temperature. Supplementary Fig. 17 shows a scenario where the ambient is cold but the covered object is hot. Just as reviewer assumed, the film is partially opened. The reason for this behavior is that the opened part in the film cools the hot object to a certain extent (the range from 6,000 to 7,200 s in Supplementary Fig. 17B). If the temperature of hot object is lower than the transition temperature of programmed 2W SMP, the film will not continue to open. A similar situation may also occur in another scenario where the ambient is hot and the covered object is cold. This is attributed to the mismatch between the transition temperature of 2W SMP and the goal of control temperature. The former is that the transition temperature is too high, and the latter is lower transition temperature.

The transition temperature of programmed 2W SMP could be modulated by the molecular weight of PCL monomers. The transition temperature increases with increasing molecular weight. This conclusion has been demonstrated in the initial submission (Supplementary Figs. 9 and 10). So, if we want to cool a hot object in a cool environment, 2W SMP synthesized by PCL monomers with low molecular weight is selected, and vice versa.

High thermal conductivity of RC tape-2W SMP film is conducive to reduce the hysteresis of the thermal management mode switch. This comment provides a new way to further improve thermal management performance of our device. Thank you.

4. *Furthermore, what is the heating/cooling power of this device in a “semi-open” state between 5 oC and 24 oC in Fig. 3b?*

Thank you for this comment. It is difficult to make an accurate definition of a “semi-open” state. To avoid misunderstanding, we refer to the intermediate morphology between heating and cooling modes as transient state. During the device switching from heating mode to a transient state (approximately half of the nano-Cr black Al plate covered by unfolding RC tape-2W SMP film) and then to cooling mode, the heat flux of the device turns from positive to negative. The real-time curves of heat flux and corresponding meteorological data have been added to the revised SI.

(SI) Page 39, Line 491-497:

Supplementary Fig. 24 | Time-resolved curves of heat flux (Φ_q) (black line) and temperature difference between dual-mode device and Al foil ($\Delta T = T_{\text{dual-mode device}} - T_{\text{Al foil}}$) (red line). The device evolves from heating mode to intermediate state, then to cooling mode. The corresponding meteorological data, including intensity of solar radiation (orange line), humidity (light-blue line) and wind speed (light-green line), are also listed.

5. On page 9, the authors wrote “the scattering center wavelength shows a blue-shift trend with the increase in diameter of TiO₂ NPs”. Is it blue-shift or red-shift?

Thank you for the correction. The sentence has been modified.

(Manuscript) Page 9, Line 212-214:

On the other hand, the scattering center wavelength shows a red-shift trend with the increase in diameter of TiO₂ NPs (Fig. 2d).

6. On page 10, the authors claimed that “The optimized RC tape can reflect more than 90% of solar radiation...”, but why this reflection becomes 85% in Fig. 2e?

Thank you for pointing this out. The optimal spectral selectivity of RC tape (90% solar reflectance and 96% mid-infrared emissivity) is achieved, when the volume fraction of TiO₂ NPs is 50% and the thickness of RC tape is 225 μm . However, the switch between thermal management modes is a morphological evolution. And its core mechanism of actuating is to minimize the internal stress at the interface between the radiative cooling layer and the actuating layer. Too thick RC tape will limit its lateral bending due to large moment of inertia, further breaking the coiled shape of RC tape-2W SMP laminate at low temperature. Therefore, 75- μm -thick RC tape is selected to fabricate dual-mode device to ensure the coiled shape of

laminate at the cost of reducing a little solar reflection. As a result, the solar reflection of dual-mode device at heating mode is 85%, rather than the optimal 90%. To avoid unwelcome confusion, the following changes are added according to the above discussion.

(Manuscript) Page 10, Line 228-230:

For the device in cooling mode, ~85% of reflection for solar radiation and ~97% of mid-infrared emission in the wavelength range of 8-13 μm are achieved (Supplementary Fig. 4).

(SI) Page 19, Line 340-344:

Even so, 75- μm -thick RC tape with TiO_2 NPs of 30% volume fraction is selected to fabricate dual-mode device to ensure the coiled shape of laminate at the cost of reducing a little solar reflection. The reason is that too thick RC tape will limit its lateral bending due to large moment of inertia, further breaking the coiled shape of RC tape-2W SMP laminate at low temperature.

7. *I feel confused by Fig. 4f as well as the supplementary Fig. 22b, where the heating and cooling modes (denoted by the color of the curve) seem to be dependent on the ON/OFF of Joule heating rather than the actual temperatures? Whenever the authors switched ON or OFF the Joule heating power, the definition of heating/cooling mode changes “instantly” without any delay? I assume that a temperature change should occur first until reaching the “transition temperature”, and then the device working mode changes accordingly. Time-dependent temperatures of both this device and the copper plate are necessary to avoid confusion.*

We agree with the remark of the reviewer. The thermal management mode of the device is dependent on its own temperature, rather than on-off of Joule heating. It is almost impossible that the temperature in a clear daytime fluctuates periodically and the fluctuation is high enough to trigger the switch of thermal management modes. As a proof of concept, we use temperature of covered object, instead of the ambient temperature, triggering the switch of thermal management modes. The temperature of the covered object is controlled by on-off of Joule heating.

The assumption that “a temperature change should occur first until reaching the ‘transition temperature’, and then the device working mode changes accordingly” is right. To avoid confusion, we re-plotted Fig. 4f, supplementary Figs. 27 and 28 (Supplementary Fig. 22 in the initial submission), and added time-resolved temperature of dual-mode device and Cu plate in the revised manuscript and SI.

(Manuscript) Page 14, Line 322:

(Manuscript) Page15, Line 338-342:

f, Real-time temperature difference ($\Delta T = T_{\text{sample}} - T_{\text{Cu plate}}$) of dual-mode device (black line) compared with 200- μm -thick Cu plate (blue dash line) under solar radiation (orange line). As Joule heating power is repeated to be on-off, the dual-mode device switches between cooling mode and heating mode by perceiving temperature.

(SI) Page 42, Line 516-526:

Supplementary Fig. 27 | Demonstration of temperature control by dual-mode device in a clear daytime. a, Schematic of temperature measurement setup for demonstration of thermal management. b, Time-resolved temperature curves of dual-mode device (black line) and 200- μm -thick Cu plate (blue dash line), along with corresponding solar radiation (orange line). As Joule heating power is repeated to be on-off, the dual-mode device switches between cooling

mode and heating mode by perceiving temperature. The real-time temperature difference between them is shown in Fig. 4f. **c**, Time-resolved curves of meteorological data, including intensity of solar radiation (orange line), humidity (light-blue line) and wind speed (light-green line), and Joule heating power (black line).

(SI) Page 43, Line 528-536:

Supplementary Fig. 28 | Demonstration of temperature control by dual-mode device in a clear nighttime. **a**, Real-time temperature and corresponding temperature difference ($\Delta T = T_{\text{sample}} - T_{\text{Cu plate}}$) of dual-mode device (black line) compared with 200- μm -thick Cu plate (blue dash line). As Joule heating power (red line) is repeated to be on-off, the dual-mode device switches between cooling mode and heating mode by perceiving temperature. **b**, Corresponding time-resolved curves of meteorological data, including intensity of solar radiation (orange line), humidity (light-blue line) and wind speed (light-green line).

8. What is the thickness of DOP-modified PMP layer in supplementary Fig. 3a and 3b?

Thanks for your reminder. The thickness of DOP-modified PMP layer in supplementary Fig. 3a and 3b is 75 μm . We modified the text in the revised SI accordingly.

(SI) Page 19, Line 335:

The thickness of RC tape is 75 μm .

Reviewer #2 (Remarks to the Author):

In this paper, a dual-mode radiation cooling / absorption heating structure is designed based on the deformation maladjusted bilayer film. The design idea of the structure is novel and can overcome the dilemma of radiation cooling materials in the low-temperature environment. However, some details of the experimental design in this paper need to be explained again:

1. The outdoor test takes copper sheet as a reference, which is very different from the actual situation. The copper sheet will absorb heat and heat up under the irradiation of sunlight, but the atmospheric temperature does not reach the same temperature at this time. In this case, the heat absorption power and radiation power of the dual-mode device are unreasonable, which will lead to underestimate the heat absorption power and overestimate the radiation power.

Thank you for this comment. This issue needs clarification. The reference for estimating solar heating power and radiative cooling power in outdoor test is Al foil, rather than bare Cu plate. That is exactly what we are considering that the bare Cu plate will absorb heat and heat up under the solar radiation. On the contrary, the temperature of Al foil is close to the ambient temperature due to its near-zero solar absorption and infrared emission. ((Manuscript) Page 15, Line 348-350: “One copper (Cu) plate is covered by our dual-mode device, and the other is covered by a same-sized aluminum (Al) foil as a control group, because its solar absorption and infrared emission are close to zero (Supplementary Fig. 13).”) Therefore, solar heating power and radiative cooling power are reasonable.

2. When radiation cooling materials can save energy consumption at high temperature, they need to be able to cool below the ambient temperature. Materials have radiation ability does not mean that they can cool below the ambient temperature. The comparison between the outdoor test temperature of the radiation film and the ambient temperature is not given in this paper; so it is impossible to judge whether it has the ability to reduce the temperature below the ambient temperature.

As reviewer pointed out, sub-ambient cooling is exactly what radiative cooling materials require to save energy at high temperatures. To demonstrate the cooling performance of RC tape in this work, an outdoor test has been performed prior to performance estimation of dual-mode device. Compared with the ambient, a ~5 K temperature drop is realized on a sunny day in a hot summer. The related experimental results have been added in the revised manuscript and SI.

(Manuscript) Page 10, Line 231-233:

Such a huge difference in the spectral characteristics of the device in the two modes lays the foundation for the zero-energy intelligent dual-mode thermal management device (Supplementary Fig. 5).

(SI) Page 20, Line 346-354:

Supplementary Fig. 5 | Time-resolved curves of meteorological data, including intensity of solar radiation (orange line), humidity (light-blue line) and wind speed (light-green line), and temperature difference (blue line, $\Delta T = T_{RC\ tape} - T_{amb}$) between RC tape ($T_{RC\ tape}$) and the ambient (T_{amb}). An average temperature drop of ~ 5 K is achieved on a clear sunny day in summer, when RC tape exposes to the sky without any convection shield in a large open square (inset images). Sub-ambient cooling is a necessary performance for radiative cooling materials to save energy at high temperature.

3. *Aluminum plate and copper plate are selected as references for indoor test and outdoor test respectively, because they have enough low thermal radiation capacity. However, if there is oxidation on the surface of aluminum and copper plates, their radiation abilities are often obvious. The infrared radiation spectrum of them is not given in this paper. In addition, why should we choose different materials for comparison indoors and outdoors?*

Thank you for reminding us. The measured reflectance spectra of Al foil and Cu plate have been added to the revised SI. High infrared reflection indicates that there is neglectable oxidation on the surface of Al foil and Cu plate.

(SI) Page 29, Line 399-403:

Supplementary Fig. 14 | Spectral characteristics of several reference materials. a,b, Reflectance spectra of (a) 20- μm -thick Al foil and (b) 200- μm -thick Cu plate. **c,d,** Absorption/emission spectra of (c) 500- μm -thick wood chip and (d) 300- μm -thick cotton. The insets are the corresponding optical images.

The reference for measuring heat flux is always Al foil, whether indoors and outdoors, as it has near-zero solar absorption and infrared emission.

The demonstration of temperature control shows the expansion of temperature changing zone, when the device switches between heating and cooling modes. Both Al foil and Cu plate have near-zero infrared emission, while the difference of solar absorption generates a stable temperature difference under fixed solar radiation. Thus, the selection of Al foil or Cu plate as reference only determines the relative temperature rise and drop of dual-mode device, and has little effect on its total temperature fluctuation.

To get closer to reality, we compared the thermal performance of the dual-mode device with four common materials with fixed spectral characteristics (including Al foil, Cu plate, wood chip and cotton). The related results have been added in the revised manuscript and SI.

(SI) Page 44, Line 538-547:

Supplementary Fig. 29 | Expansion of temperature change zone by dual-mode device. a, Time-resolved curves of meteorological data, including intensity of solar radiation (orange line), humidity (light-blue line) and wind speed (light-green line). **b,** Temperature change zone of dual-mode device and four common materials with fixed spectral characteristics (corresponding to Supplementary Fig. 14). The average temperature of Al foil is used as reference to calculate temperature change ($\Delta T = T_{\text{sample}} - T_{\text{Al foil}}$). Al foil, gray bar; Cu plate, orange bar; wood chip, green bar; cotton, purple bar; dual-mode device, color-graduated bar (heating mode is red boundary and cooling mode is blue boundary). The error bars represent the measuring standard deviation.

4. *The heat absorption power and radiation power of dual-mode device are evaluated by the input power of heater and thermoelectric device. Does it not need to consider the working efficiency of these devices? Especially the efficiency of thermoelectric refrigeration.*

Thanks for your consideration. Because the heater is a pure resistor load, the input power is calculated by Joule's law ($P = I^2 R$, where P is the power converted from electrical energy to thermal energy, I is the current through the heater, and R is the resistance of the heater). Compared with the heater, the thermoelectric device is not a pure resistor load and its heating or cooling power can not be calculated by Joule's law, so the thermoelectric device is just used as a constant cold source to avoid calculation of efficiency of thermoelectric refrigeration. In

doing so, the heating and cooling power is only equal to the difference of the input power of heater between the device and the reference (Al foil). The detail about the experimental method has been described in the initial submission ((SI) Page 5, Line 96-101: “The heater in the apparatus for dual-mode device was connected with a constant current source, and the other was connected with a feedback-controlled heating system to always maintain the temperature of Al foil at the same temperature as the dual-mode device. Joule heating power applied on Al foil minus that on dual-mode device was defined as solar heating power (positive heat flux) or radiative cooling power (negative heat flux).”)

5. *Theoretical calculations are based on Formula 1, but it is not clear how to use some key parameters. For example, when calculating indoor heat absorption and radiation power, what is the used environmental radiation spectrum?*

Thank you for the suggestion. For clearly understanding analysis of radiative heat loss and solar absorption, the key parameters are explained in detail.

(SI) Page 11, Line 228-245:

In the indoor environment, the device is surrounded by a roof, walls and some other objects, which have a combined infrared emissivity close to 100% in general. Considering this situation, therefore, the overall indoor environment can be considered as a room-temperature (T_{amb}) thermal radiative source. This simplifies Supplementary Equation (3) into

$$P_{atm}(T_{amb}) = A_{device} \int_0^{\infty} d\lambda I_{BB}(T_{amb}, \lambda) \varepsilon_{device}(\lambda)$$
. In the outdoor environment, the existence of outer space at low temperature makes dual-mode devices have radiative heat loss by reducing the input of infrared radiation from the ambient to the device. Radiative heat absorption of the device from the ambient is only from the atmosphere, while the radiative radiation from out space is negligible due to its extremely cold temperature. In this scene, Supplementary

Equation (3) is reducible to
$$P_{atm}(T_{amb}) = A_{device} \int_0^{\infty} d\lambda I_{BB}(T_{amb}, \lambda) \varepsilon_{device}(\lambda) (1 - t_{atm}(\lambda))$$
, where $t_{atm}(\lambda)$ is the atmospheric transmissivity in the zenith direction.

The input solar power is supplied by a simulated solar source with a spectral distribution of AM 1.5. In the dark (no simulated sunlight), the solar radiation is zero, so the absorption of device from solar radiation (P_{sun}) is also zero. And under the simulated sunlight, $I_{sun}(\lambda)$ is replaced by the spectral distribution of AM 1.5, which is used to calculate the absorption of device from solar radiation (P_{sun}) by Supplementary Equation (4).

(SI) Page 35, Line 450-452:

The ambient temperature (T_{amb}) is fixed as 29 °C, and the combined non-radiative heat coefficient (h_c) is assumed as 0.

Reviewer #3 (Remarks to the Author):

The authors developed an innovative way to achieve an auto-switched and zero-energy dual-mode radiative thermal management device and demonstrated the function through folding and unfolding the laminate structure to the variation of ambient temperature. The idea is interesting but several questions should be addressed before making decision.

1. The results in this work only show the temperature reduction close to 15 K in cooling mode and growth close to 6 K in the heating mode by just comparing with a bare Cu plate. Therefore, the description in abstract 'A practical demonstration shows that the temperature fluctuation is reduced by ~21 K.' should be specified by comparing with a bare Cu plate. However, photonic devices designed for thermal management are generally used in outdoor structures, such as buildings, cars, and space aerocrafts, as well as soft and wearable textiles, etc. Hence, I think the authors should give a little more comparison of the thermal performance with, for instance, two or three other materials.

Thank you for the suggestion. The description in abstract has been revised.

(Manuscript) Page 1, Line 15-16:

A practical demonstration shows that the temperature fluctuation is reduced by ~21 K, compared with copper plate.

To get closer to reality, we compared the thermal performance of dual-mode device with four common materials with fixed spectral characteristics (including Al foil, Cu plate, wood chip and cotton). The related results have been added in the revised manuscript and SI.

(SI) Page 44, Line 538-547:

Supplementary Fig. 29 | Expansion of temperature change zone by dual-mode device. a, Time-resolved curves of meteorological data, including intensity of solar radiation (orange line), humidity (light-blue line) and wind speed (light-green line). **b,** Temperature change zone of dual-mode device and four common materials with fixed spectral characteristics (corresponding to Supplementary Fig. 14). The average temperature of Al foil is used as reference to calculate temperature change ($\Delta T = T_{\text{sample}} - T_{\text{Al foil}}$). Al foil, gray bar; Cu plate, orange bar; wood chip, green bar; cotton, purple bar; dual-mode device, color-graduated bar (heating mode is red boundary and cooling mode is blue boundary). The error bars represent the measuring standard deviation.

2. The authors stated that the average solar heating power and radiative cooling power are $\sim 859.8 \text{ W/m}^2$ and $\sim 126.0 \text{ W/m}^2$ in heating and cooling modes, respectively, which correspond to folding and unfolding of the laminate. My question is, is the result from all of the area in unfolding state? While in folding laminate, especially in the case of film array, both in horizontal and vertical direction, only a part of the area will contribute to the heating effect because the other part of the area will be covered by the coiled laminate. As such, the values of power of thermal load of heating mode should be revised.

Thank you for this comment. The results of average solar heating power and radiative cooling power are actually from all of the area in unfolding state, namely $40 \text{ mm} \times 40 \text{ mm}$. To

maximize the solar-thermal conversion of the device per area, the coiled RC tape-2W SMP film is set on the northern edge of nano-Cr black Al plate (relative position noted in Supplementary Fig. S22a). In addition, there is a relatively small vertical angle of solar radiation in autumn. It reduces the shielding of sunlight by the coiled RC tape-2W SMP film, further increasing the heated area of nano-Cr black Al plate, when the device works in heating mode. The effect of incident angle of the solar radiation is discussed in response to Comment #4 in detail.

As reviewer pointed out, either horizontally or vertically, only a part of area could contribute to the heating effect in the case of device array. The percentage improvement of active area in the device could also alleviate this issue. The influence of device size on the effectivity of thermal loading is discussed in response to Comment #3 in detail. The experiment about the measurement of solar heating power and cooling power reported in the initial submission is just performed on a single device, rather than device array.

3. Since the present intelligent dual-mode devices are the folding and unfolding laminates, I think the size of a single plate will affect the effectivity of thermal loading, switching, and potential applications of the devices. Therefore, additional results from the devices with different sizes maybe more interesting.

We thank the reviewer for this comment. The size of a single plate is indeed an important factor in the effectiveness of thermal loading, switching, and potential applications of the devices.

Without considering gravity, the radius of curvature of RC tape-2W SMP laminate is length-independent, which depends on the difference of length between RC tape and 2W SMP at different temperatures (similar to the mismatch of thermal expansion coefficients in some typical thermal actuators). It means that the longer RC tape-2W SMP laminate, the higher ratio of active area in the device. This device would have an effectiveness of thermal loading close to the prediction.

In the real world, gravity is not a neglectable factor for the morphological evolution of RC tape-2W SMP laminate, especially if it is very long. In this case, the radius of curvature becomes larger with the increase of gravity, and RC tape-2W SMP film can not even completely roll up in extreme conditions. Instead of increasing, the effectivity of thermal loading of device falls dramatically.

The size mainly determines the ratio of active area of the device in heating mode. We made a comparison of heating performance of the devices in heating mode with different sizes. This comparison is strong evidence that shows larger device has a better heating performance, when RC tape-2W SMP films have the same radius of curvature. The results and related discussion have been added in the revised manuscript and SI.

(Manuscript) Page 12, Line 290-292:

Then, several RC tape-2W SMP laminates are placed side by side and bonded together to form a large-sized film exactly covering the nano-Cr black Al plate (Supplementary Note 2 and Supplementary Figure S21).

(SI) Page 8, Line 142-166:

Supplementary Note 2. Influence of size of dual-mode device on effectiveness of thermal

loading

The size of dual-mode device is indeed an important factor in the effectiveness of thermal loading, switching, and potential applications of the devices. The nature of the issue is the influence of size on the morphological evolution of RC tape-2W SMP film.

Without considering gravity, the radius of curvature of RC tape-2W SMP laminate is length-independent, and depends on the difference of length between RC tape and 2W SMP at different temperatures. It means that the longer RC tape-2W SMP laminate, the higher ratio of active area in the device. The device with large size would have an effectiveness of thermal loading close to the prediction. But in the real world, gravity is not a neglectable factor for the morphological evolution of RC tape-2W SMP laminate, especially if it is very long. In this case, the radius of curvature becomes larger with the increase of gravity. RC tape-2W SMP film can not even completely roll up in extreme conditions. Instead of increasing, the effectiveness of thermal loading of device falls dramatically.

A study on the temperature of the device in heating mode with different sizes is done, thereby visually showing the influence of the size on the heating performance (Supplementary Fig. 21). The radius of curvature is almost the same, when RC tape-2W SMP films are within the limited length (from 10 mm to 40 mm in this experiment). The coiled RC tape-2W SMP films with the same radius of curvature will shade the same area of nano-Cr black Al plate. In other words, the device with the larger size has the higher solar heating power close to the prediction. As one might expect, the temperature rise of the device gradually decreases with the shrinking of active area in the device. The size of the device in next experiments is fixed at 40 mm × 40 mm to achieve an efficient thermal management performance.

(SI) Page 36, Line 461-471:

Supplementary Fig. 21 | Influence of size of dual-mode device on effectiveness of thermal loading. Time-resolved temperature curves of dual-mode device in heating mode with different sizes, along with the corresponding solar radiation. It should be noted that all RC tape-2W SMP films are locked tightly to maintain a coiled state. The reason is that unconstrained film will completely unfold and the device works at cooling mode in the hot summer. The coiled RC tape-2W SMP film will shield a part of nano-Cr black Al plate, when the device works in heating mode. For a smaller device, especially the shorten of the length along the unfolding direction of the laminate, the reduction of the active zone in nano-Cr black Al plate decreases

the total heating power per area of the device.

4. *The effect of incident angle of solar radiation should be discussed, especially in the cases that the size of laminates is not much smaller, because it will greatly affect the absorption.*

Thank you for the insightful suggestion. We agree with the reviewer's comment that incident angle of solar radiation will greatly affect the absorption, especially if the size of laminates is not much smaller. In the initial submitted version (Fig. 4a and Supplementary Fig. 23), all of the data in field test are achieved when RC tape-2W SMP film is set on the northern edge of nano-Cr black Al plate (relative position noted in Supplementary Fig. S22a). This design maximizes the solar absorption of dual-mode device under heating mode, considering the relative position between the dual-mode device and the sun. To address this issue, we have conducted a new series of experiments with the data shown below and added to the revised SI (Supplementary Fig. S22b), where four RC tape-2W SMP films are set on the different edges of nano-Cr black Al plate in four individual devices. As expected, the device with the film on the northern edge often achieves the maximum temperature rise, while the device with the film on the southern edge has the minimum temperature rise. When the sun rises in the east, the device with the film on the western edge is hotter than that with the film on the eastern edge. And this temperature relationship switches to the opposite when the sun sets in the west. It should be noted that all RC tape-2W SMP films are locked to maintain a coiled state. The reason is that unconstrained film will completely unfold and the device works at cooling mode in the hot summer.

(Manuscript) Page 15, Line 344-347:

Further, we performed a daily field experiment in the real-world scenarios outdoors to test its truly practical thermal management performance in a real environment (located on the campus of Nankai University in Tianjin (38.99N, 117.34E), China) (Supplementary Note 5).

(SI) Page 13-14, Line 258-287:

Supplementary Note 5. Effect of incident angle of solar radiation on solar-thermal conversion

In this design, the switch of thermal management modes is realized by morphological evolution of RC tape-2W SMP film. For heating mode, the film rolls to one side to maximize the exposed area of the nano-Cr black Al plate. And for cooling mode, the film spreads out until completely covering the bottom nano-Cr black Al plate. The device in cooling mode is almost isotropic in two-dimensional working plane. By contrast, when the device works in heating mode, the coiled film shields a part of nano-Cr black Al plate away from the solar radiation. This will lower the solar-thermal conversion efficiency of the device per area. Thus, it is very important to carefully select the relative position between the coiled film and the incident sunlight to improve solar-thermal conversion efficiency of the device in heating mode.

To address this issue, we conducted a set of parallel experiments, where four coiled RC tape-2W SMP films are set on the different edges of nano-Cr black Al plate in four individual devices (**Supplementary Figure 22**). Because the testing position is in the northern hemisphere (Tianjin: 38.99N, 117.34E), the sun is south of the device for much of the daytime, especially in winter. As expected, the device with the film on the northern edge achieves the maximum temperature rise, while the device with the film on the southern edge has the minimum

temperature rise. When the sun rises in the east, the device with the film on the western edge is hotter than that with the film on the eastern edge. And this temperature relationship switches to the opposite when the sun sets in the west. It should be noted that all RC tape-2W SMP films are locked tightly to maintain a coiled state. The reason is that unconstrained film will completely unfold and the device works at cooling mode in the hot summer. Overall, when the coiled RC tape-2W SMP film is set on the edge opposite to the sun, the device in heating mode could achieve the maximum solar-thermal conversion efficiency without negative effect on cooling ability of the device in cooling mode (namely set on the northern edge when in the northern hemisphere and the southern edge when in the southern hemisphere). Accordingly, all of the data in field test are achieved when RC tape-2W SMP film is set on the northern edge.

(SI) Page 37, Line 473-482:

Supplementary Fig. 22 | Effect of incident angle of solar radiation on solar-thermal conversion. **a**, Schematic illustration showing four coiled RC tape-2W SMP films separately set on the different edges of nano-Cr black Al plate in four individual dual-mode devices. Because the testing position is around latitude 39 degrees north, the sun is often south of the device, especially at noon. **b**, Time-resolved curves of intensity of solar radiation (orange line) and temperature of dual-mode devices in heating mode. The coiled RC tape-2W SMP films are set on the different edges of nano-Cr black Al plate in four individual devices: on the northern edge (black line), on the eastern edge (red line), on the western edge (blue line), and on the southern edge (green line).

5. Additionally, the authors also reported the other triggered temperature 'This RC tape-2W SMP laminate was coiled at room temperature and unfolded at 55 °C.' In fact, such triggered temperature is somewhat a little higher value in most cases. Can the devices also work as the ambient temperature is around 45 °C? 50 °C?

It is true that the triggering temperature of RC tape-2W SMP laminate should match the requirement of temperature control in application scenario well. In the initial submission, we have demonstrated that the increase of molecular weight of PCL monomers (or PCL-diol monomers) will raise the phase-changing temperature of programmed 2W SMP (Supplementary Fig. S8), further increasing the triggered temperature of RC tape-2W SMP laminate (Fig. 3B and Supplementary Fig. S9). Based on this, we could select the 2W SMP synthesized by PCL monomers with appropriate molecular weight (between 10,000 and 36,000), according to the requirements of triggered temperature in the actual cases. Thus, the goal that the devices work as the ambient temperature is around 45 °C or 50 °C, can come true.

(Manuscript) Page 11, Line 252:

Fig. 3b

(SI) Page 24, Line 369:

Supplementary Fig. 9

(SI) Page 25, Line 380:

Supplementary Fig. 10

6. Can the devices also work well as the substrate becomes to, for instance, soft or wearable textiles?

It is an interesting idea. We have demonstrated that the morphology of RC tape-2W SMP film depends on its own temperature in nature, rather than the mechanical properties of the substrate. Thus, RC tape-2W SMP film is fully able to realize coiled-unfolded morphological evolution, when the substrate becomes to soft or wearable textiles (a little similar to Hsu's study: moisture-responsive actuator for personal thermal management (*Sci. Adv.* 7, eabj7906 (2021))). Even so, how to improve the thermal contact between RC tape-2W SMP film and soft or wearable textile is a key issue that has to be solved, for fully utilization of generated heating or cooling energy. However, this idea motivates us to do more investigations in the future study to optimize the structures of dual-mode device, thus expanding applications.

Reviewer #4 (Remarks to the Author):

The paper introduces a high-performance, intelligent auto-switched and zero energy dual-mode radiative thermal management device. It describes in details the novel features of the design. It demonstrates its performance via modelling, lab tests and in real-life outdoor settings.

The provided evidences support the conclusions and the novel technology sounds interesting and with potential.

The authors build the argumentation behind the novel technology on “the fact of the global total primary energy demand is close to a 15-billion-tonne oil equivalent in 2019, and nearly 50% of energy consumption is merely used for daily heating and cooling (...) Therefore, it is particularly important and imperative to develop various feasible high-performance thermal management technologies with low or even zero energy consumption, which is able to reduce fossil energy demand and further emission of greenhouse gases” and presents its heating and cooling potential for various locations. This is very right. Yet, the conclusions lack a section describing the potential application of this new device. It would significantly enhance the value of the paper if suggestions, for how and where this technology can be applied, were part of the paper.

I hope these suggestions will improve your manuscript and make it even more valuable to the field.

Thanks for your suggestion. We have written a text about the potential applications of dual-mode device in the revised manuscript.

(Manuscript) Page 18-19, Line 440-454:

As a zero-energy design, the dual-mode thermal management device takes full advantage of renewable energy in nature, solar heat and space cold, thereby very well-suitable for open areas, such as roofs of large-scale buildings. The generated thermal energy could be used as a direct source for space temperature control by arranging a large number of devices on the roof, but this is a low-efficiency selection and just able to regulate temperature where the space is close to the roof. Referring to the reported studies, the application of radiative thermal management to initial temperature control of heat-transfer fluid in active thermal management systems will significantly improve efficiency in thermal energy use. In general, this idea represents a system-level approach to renewable energy generation and efficiency in the future. As far as an individual device is concerned, how to further improve its thermal management performance and weather resistance in the real scenario is the top priority. However, such a real zero-energy dual-mode device would have great and practical potential for global thermal management and energy saving, and provides a renewable zero-energy platform to realize the goal of Net Zero Carbon 2050.

REVIEWER COMMENTS

Reviewer #1 (Remarks to the Author):

The authors have properly addressed my questions and I do not have further comments on this manuscript.

Reviewer #2 (Remarks to the Author):

The author replied to the comments of the reviewers and made corresponding amendments to the paper. But there is still one question: the author thinks that the near zero absorption of sunlight by aluminum makes it have the same temperature as the ambient temperature. Firstly, although the absorption of aluminum to the sun is very low, it exceeds 5%. The power density of absorbing sunlight can reach 40-50 watts per square meter or higher. The author concludes that the temperature of the aluminum plate is the same as the ambient temperature. Where is the ambient temperature test probe on the test device?

Reviewer #3 (Remarks to the Author):

The authors have addressed all my questions. I therefore recommend it for publication now.

Dear Reviewers,

We greatly appreciate your thoughtful comments. Herewith we are submitting the revised manuscript entitled, “Temperature-Dependent Dual-Mode Thermal Management Device with Net Zero Energy for Year-Round Energy Saving” for publication as an Article in *Nature Communications*.

The list of changes starts on the next page. Additional discussion have been added to address the comments of reviewers. For clarity, the reviewers’ comments are written in *italics* below, and our responses to the reviewers are in **dark red** while the modifications to the manuscript and SI are in **blue**.

Thank you very much for your time.

Sincerely,

Professor xxx

Reviewer Comments

Reviewer #2 (Remarks to the Author):

1. The author replied to the comments of the reviewers and made corresponding amendments to the paper. But there is still one question: the author thinks that the near zero absorption of sunlight by aluminum makes it have the same temperature as the ambient temperature. Firstly, although the absorption of aluminum to the sun is very low, it exceeds 5%. The power density of absorbing sunlight can reach 40-50 watts per square meter or higher. The author concludes that the temperature of the aluminum plate is the same as the ambient temperature. Where is the ambient temperature test probe on the test device?

Thanks for this comment. In the manuscript and SI, we did not conclude that the temperature of Al foil was the same as the ambient temperature. The device shown in Supplementary Fig. S18 is used to estimate solar heating power and radiative cooling power of dual-mode device, rather than the difference of temperature between the device and the ambient. ((SI) Page 5, Line 96-99: “The heater in the apparatus for dual-mode device was connected with a constant current source, and the other was connected with a feedback-controlled heating system to always maintain the temperature of Al foil at the same temperature as the dual-mode device.” (SI) Page 5, Line 106-111: “For solar heating, Peltier devices and fans combined with them operated under the same conditions to maintain a lower temperature, which endows the dual-mode device in heating mode. For radiative cooling, Peltier devices and fans were still in the working state. By the heater, an appropriate constant Joule heating power was applied on the dual-mode device to make the device switch to cooling mode.”) So, there is no thermocouple to record the ambient temperature.

As the reviewer claimed, Al foil does absorb a little solar radiation (the weighted-average solar absorptivity of ~6.5% estimated from simulated spectrum (Supplementary Fig. 13)). Therefore, this part of solar absorption should be calibrated when Al foil is chosen as the reference in the field test. Solar heating power and radiative cooling power shown in Fig. 4a have been calibrated in the initial submission. To avoid confusion, we have rewritten the related text on data analysis in the revised SI.

(SI) Page 5, Line 99-106:

Because Al foil has a little solar absorption (the weighted-average solar absorptivity of ~6.5% estimated from simulated spectrum (Supplementary Fig. 13)), the heat flux (Φ_q) should be calibrated based on this additional solar-thermal conversion. Solar heating power (positive heat flux) or radiative cooling power (negative heat flux) is calculated by $\Phi_q = P_{Al} - P_{device} + \alpha_{Al} I_{sun}$, where P_{Al} is Joule heating power per area applied on Al foil, P_{device} is Joule heating power per area applied on device, α_{Al} is weighted-average solar absorptivity of Al foil and I_{sun} is the intensity of solar radiation.

REVIEWERS' COMMENTS

Reviewer #2 (Remarks to the Author):

The authors have properly addressed my questions and I do not have further comments on this manuscript.